# Transcriptome Analysis and miRNA Target Profiling at Various Stages of Root-Knot Nematode *Meloidogyne incognita* Development for Identification of Potential Regulatory Networks

**DOI:** 10.3390/ijms22147442

**Published:** 2021-07-12

**Authors:** Vimalraj Mani, Awraris Derbie Assefa, Bum-Soo Hahn

**Affiliations:** 1Department of Agricultural Biotechnology, National Institute of Agricultural Sciences, Rural Development Administration, Jeonju 54874, Korea; vimalraj08@gmail.com; 2National Agrobiodiversity Center, National Institute of Agricultural Sciences, Rural Development Administration, Jeonju 54874, Korea; awraris@korea.kr

**Keywords:** plant-parasitic nematode, *Meloidogyne incognita*, RNA sequencing, miRNA, target prediction

## Abstract

Root-knot nematodes (RKNs) are a group of plant-parasitic nematodes that cause damage to various plant species and extensive economical losses. In this study, we performed integrated analysis of miRNA and mRNA expression data to explore the regulation of miRNA and mRNA in RKNs. In particular, we aimed to elucidate the mRNA targets of *Meloidogyne incognita* miRNAs and variations of the RKN transcriptome during five stages of its life cycle. Stage-wise RNA sequencing of *M. incognita* resulted in clean read numbers of 56,902,902, 50,762,456, 40,968,532, 47,309,223, and 51,730,234 for the egg, J2, J3, J4, and female stages, respectively. Overall, stage-dependent mRNA sequencing revealed that 17,423 genes were expressed in the transcriptome of *M. incognita*. The egg stage showed the maximum number of transcripts, and 12,803 gene transcripts were expressed in all stages. Functional Gene Ontology (GO) analysis resulted in three main GO classes: biological process, cellular components, and molecular function; the detected sequences were longer than sequences in the reference genome. Stage-wise selected fragments per kilobase of transcript per million mapped reads (FPKM) values of the top 10 stage-specific and common mRNAs were used to construct expression profiles, and 20 mRNAs were validated through quantitative real-time PCR (qRT-PCR). Next, we used three target prediction programs (miRanda, RNAhybrid, and PITA) to obtain 2431 potential target miRNA genes in RKNs, which regulate 8331 mRNAs. The predicted potential targets of miRNA were generally involved in cellular and metabolic processes, binding of molecules in the cell, membranes, and organelles. Stage-wise miRNA target analysis revealed that the egg stage contains heat shock proteins, transcriptional factors, and DNA repair proteins, whereas J2 includes DNA replication, heat shock, and ubiquitin-conjugating pathway-related proteins; the J3 and J4 stages are represented by the major sperm protein domain and translation-related proteins, respectively. In the female stage, we found proteins related to the maintenance of molybdopterin-binding domain-containing proteins and ubiquitin-mediated protein degradation. In total, 29 highly expressed stage-specific mRNA-targeting miRNAs were analyzed using qRT-PCR to validate the sequence analysis data. Overall, our findings provide new insights into the molecular mechanisms occurring at various developmental stages of the RKN life cycle, thus aiding in the identification of potential control strategies.

## 1. Introduction

Plant-parasitic nematodes are the most destructive group of plant pathogens. Plant-parasitic nematodes are extremely challenging to control and cause global agricultural losses of approximately USD 157 billion annually. *Meloidogyne incognita* is widely distributed in temperate and tropical regions worldwide; it can infect thousands of plant species, including almost all vegetable crops [1]. The life cycle of this nematode is divided into five stages: egg, three juvenile stages (infective stage J2 and developmental stages J3 and J4), and the female (or male) adult stage. The saccate female remains within the root and produces eggs [2]. Because *M. incognita* can affect a wide range of hosts and cause extensive damage, the development of effective strategies to control this pest is essential. Molecular studies concerning *M. incognita* can shed light on its physiology and growth, thereby providing potential environmentally friendly modes of control to replace widely used hazardous nematicides.

Early large-scale molecular studies of the root-knot nematode *M. incognita* have produced genome sequence data [2] and microRNA (miRNA) profiles [3,4]. miRNAs are a diverse class of small non-coding endogenous RNAs with lengths ranging from 16 to 29 nucleotides (nt) and a mean length of 22 nt [5]. Thus far, thousands of miRNAs have been identified in several unicellular and multicellular eukaryotes, including humans, insects, nematodes, and plants. Since the discovery of the first miRNA, lin 4 in *Caenorhabditis elegans* [6], many miRNA sequences have been deposited in miRBase, a publicly accessible database. The accumulated evidence shows that miRNAs play major roles in almost all biological and metabolic processes, including reproduction, growth, organ differentiation, responses to stressors, and disease pathogenesis [7]. These reports serve as valuable sources of information to support future research regarding root-knot nematodes. However, these databases contain genomic data, and bioinformatics pipelines do not measure expression; therefore, experimental verification is needed for each gene or miRNA of interest.

Transcriptome profiling or RNA sequencing can be used to create a database of expressed cellular transcripts, which reflect the active processes occurring in the cell. This method offers a progressive step toward the development of genetic resources for root-knot nematode research. Transcriptome-based studies have been conducted for *Meloidogyne enterolobi*, *Globodera pallida*, and *Pratylenchus coffeae*, among plant-parasitic nematodes, as well as *Haemonchus contortus* and *Ancyclostoma caninum* among animal nematodes present in plant structures such as syncytia [8,9,10,11,12]. Few RNA-based studies have been conducted for *M. incognita* or plant tissues infected with it [13,14,15,16]. These studies illustrate the expressed sequence tags from infective juveniles (J2) of *M. incognita*, the difference in transcripts between the J2 and J3 stages associated with oxidative stress, esophageal gland secretions related to pathogenicity, and the role of resistance genes in plants in controlling root-knot nematode infection. Tanake et al. reported transcriptional profiles based on stage-specific RNA sequence data from the pine wood nematode *Bursaphelenchu xylophilus* [17], while Bagnaresi et al. scrutinized the transcriptome profile of *Solanum torvum* inoculated with the nematode *M. incognita* using 454 pyrosequencing and microarray technologies [18]. Shukla et al. identified gene expression profiles and metabolic networks in response to nematode growth on tomato [19]. To our knowledge, there have been few studies regarding the mechanism through which miRNA targets mRNA in *M. incognita*. 

New molecular research involving large-scale analysis of the *M. incognita* transcriptome, previously available data regarding root-knot nematodes, and stage-wise mRNA profiles would help to elucidate the biochemical changes that occur during nematode development. Therefore, in the present study, we aimed to identify the targets of miRNAs (known and novel) reported in our previous study [20], which elucidated the stage-dependent expression of mRNAs in *M. incognita*. Comprehensive transcriptomic profiles across the development stages of *M. incognita* were obtained and microRNAs targeting mRNAs were identified using various bioinformatics pipelines; expression levels were then validated through quantitative real-time PCR (qRT-PCR) analysis.

## 2. Results

### 2.1. Stage-Wise Transcriptome Profiling of M. incognita

To acquire high-throughput transcriptome data for *M. incognita*, we conducted Illumina-based next-generation sequencing. In this study, we performed RNA sequence analysis of all five developmental stages of *M. incognita*. We used three different strategies: RNA sequencing of *M. incognita* at all developmental stages, targeting of mRNA using a bioinformatics pipeline based on previously identified miRNA, and functional annotation using Blast2GO. The workflow of the current study is illustrated in Figure 1.

Deep RNA sequencing achieved the following numbers of raw reads: egg, 70,310,943; J2, 61,150,827; J3, 51,342,127; J4, 58,016,948; and female stage, 64,618,511. From the raw reads, further exclusion of non-informative sequences was conducted using CASAVA (1.8.2) and CLC genome cell (4.0). Finally, we obtained the following numbers of high-quality reads: egg, 56,902,920; J2, 50,762,456; J3, 40,968,532; J4, 47,309,223; and female stage, 51,730,234. Furthermore, we used a bioinformatics pipeline for miRNA target prediction with a reference genome. Initial preprocessing followed by mapping of sequences to the reference genome indicated that 51.31–61.12% of the high-quality sequences could be mapped to the currently available reference genome for *M. incognita* (Table 1). Through further analysis, we generated a dataset supporting transcription of 17,423 (85.55%) of the predicted gene models (Appendix A).

The stage-wise expression patterns of gene transcripts with FPKM ≥ 0.3 were assessed in the egg, J2, J3, J4 and female stages; 15,798 15,564, 14,908, 15,226, and 14,519 transcripts were identified, respectively (Table 2). The predicted numbers of genes were similar across all developmental stages.

In terms of mRNA expression, the maximum stage-specific transcript number observed was 390 in the egg stage; 12,803 gene transcripts were commonly expressed across all stages. In total, 546 gene transcripts, the highest number shared among any four stages, were expressed in the four stages other than female stage. In the J2, J3, J4, and female stages, 326, 141, 115, and 120 transcripts were exclusively expressed, respectively. The remaining gene transcripts were shared among combinations of two, three, or four stages (e.g., egg + J2, egg + J3, egg + J4, and egg + female). The greatest number of mRNAs shared between two stages was 290 for the egg and J2 stages. No mRNAs were shared exclusively between the egg and J4 or the J2 and female stages. The numbers of gene transcripts shared among various stages are presented in a Venn diagram (Figure 2). 

### 2.2. Functional Annotation and Comparison among Stages

To obtain functional annotations, we have utilized the Blast2GO tool suite—a comprehensive bioinformatics tool for functional annotation of sequencing data. BLAST2GO analysis revealed that the results of our mRNA sequencing experiment were similar to the available mRNA sequences for *M. incognita* (Project ID: PRJEA28837). We compared the top Gene Ontology (GO) distributions in three GO categories (biological process, molecular function, and cellular component) to the reference mRNA sequence, and the most common comparison sequence hits between ours and the reference study are as follows: In the biological process category, the cellular process (6058 vs. 4610), single-organism process (3645 vs. 5117), and metabolic process (4123 vs. 5301) were most strongly represented (Figure 3A). In the molecular function category, binding (5139 vs. 4368) and catalytic activity (4478 vs. 3455) were most strongly represented (Figure 3B); in the cellular component category, cell (5125 vs. 3705), organelle (3326 vs. 2619), and membrane (2569 vs. 1714) were predominant (Figure 3C). The fewest sequence hits were for electron carrier (43 vs. 29) and antioxidant activity (58 vs. 33), both in the molecular function category (Figure 3B). Comparison of enzymes encoded by the identified mRNAs indicated that similar families of enzymes were expressed, except for higher numbers of oxidoreductases (342 vs. 203) and transferases (653 vs. 565), compared with the reference mRNA sequence (Project ID: PRJEA28837). Among hydrolases, 934 enzymes were expressed in the reference study, but this number decreased to 583 in the present study. Overall, the families of enzymes expressed were similar between studies, but more transcripts were obtained in this study (Appendix A). We annotated the sequences using the available InterProScan database as a reference to identify common and specific targets. Overall, InterProScan results indicated that fewer InterProScan (516) and GO (282) data were available for our contigs, compared with the reference dataset (Appendix A). Gene functional annotation analyses of common and stage-wise expression, which are summarized in Appendix A, indicated that most of the genes were uncharacterized proteins from diverse species, including *C. elegans*, *Caenorhabditis*
*briggsae*, *Aedes*
*aegypti*, *Herpetosiphon aurantiacus*, and *Vitis vinifera*. Comparison between the reference dataset and our study showed that other functional proteins were also shared. Furthermore, the selected highly expressed mRNAs represented an uncharacterized protein, FMRF-amide-like peptide, cuticle collagen, and an uncharacterized protein in the egg, J2, J3, J4, and female stages, respectively. Highly expressed mRNAs from each stage of *M. incognita* are listed in Table 3. 

In total, 12,803 gene transcripts were commonly expressed at all stages analyzed; highly expressed common genes included a transport protein similar to SEC-2, the cell structure proteins actin and collagen (COL-1), and ribosomal and DNA-related high-mobility proteins (Table 4).

### 2.3. Heatmap Expression and Functional Analysis 

All FPKM values obtained through stage-specific analysis were subjected to two clustering processes based on the most abundant sequences and overall expression profiles (Figure 4 and Appendix A, respectively) using a heatmap. The first clustering analysis included 60 highly expressed genes, with 10 genes specific to each stage and 10 genes commonly expressed at all stages. The second clustering method, based on overall gene expression profiles at all stages, revealed grouping of mRNA from the J3 and J4 stages, which formed a subclade similar to the egg-stage mRNA profile. Overall, mRNA from the female stage was distinct from other stages of nematode development. 

### 2.4. miRNA Target Prediction and Regulation of mRNA Expression

To determine the targets of the miRNAs identified in our previous study, which reported 3107 total predicted miRNAs [3], the targets of these miRNAs were predicted from the 3′ untranslated regions (UTRs) of *M. incognita* using in silico analysis. *M. incognita* 3′ UTRs were more abundant across all ranges of sequence lengths, compared with the other nematode species *C. elegans* and *Brugia malayi*. However, for sequences 100–200 bp in length, sequences from the free-living nematode *C. elegans* were slightly more abundant. Furthermore, we performed sequence comparisons with mammals (human, mouse, and rat) and other organisms (fruit fly, zebrafish, and starlet sea anemone). Notably, except for sequences 0–100 bp in length, sequences from mammals and other groups were considerably more abundant at all sequence lengths (Figure 5, Appendix A). In *M. incognita*, a total number of 20,365 genes were predicted in the 3′ UTR region and further, we selected 17,177 genes with N (any base) percentage less than 1 and sequences longer than 10 bp (Appendix A). 

Pooling of common miRNAs and their targets was conducted using three software programs: miRanda, RNAhybrid, and PITA. Our analysis revealed 2431 miRNAs that could potentially target 8331 mRNAs. Selected stage-specific miRNA-targeted mRNA transcript IDs and hit descriptions are listed in Table 5. Analysis of stage-specific miRNA targets revealed that heat shock, DNA repair, domain-containing proteins, and transcription factors were targeted in the egg stage, whereas cell cycle regulation, DNA replication, heat shock, and ubiquitin conjugation pathway-related proteins were targeted in the J2 stage. In the J3 stage, the major sperm protein (MSP) domain was identified among the highly targeted proteins. This protein has been linked to the motility of nematode sperm, as well as signaling functions [21]. Ras-related and myocardial muscle degradation indicator proteins were also targeted in the J3 stage. In the J4 stage, targets included translation-related proteins (RNA polymerase and small ribosomal subunit 40S). In the female stage, targets included molybdopterin-binding domain and sequences associated with ubiquitin-mediated protein degradation. Regulatory proteins such as adenyl cyclases were also targeted at the female stage.

Further analysis was conducted to elucidate the interactions of miRNAs with mRNAs using the whole-genome sequence of *M. incognita* from the NCBI (ASM18041v1) and WormBase assembly (ASM18041v1a), which resulted in 1471 miRNAs (1218 known, 263 novel) targeting 7888 mRNAs. These miRNAs had a varying number of targets; one miRNA could target one to hundreds of mRNA. More than 40% of miRNAs targeted 2–10 mRNAs, while roughly 20 to 40% of miRNAs targeted 11–50 mRNAs among all stages (Figure 6).

GO analysis indicated that the reference (PRJE28837) mRNA-targeted biological processes included cellular processes, while the molecular function mRNAs were involved in binding and catalytic activity. Among cellular components, mRNA targets included the organelle and cell classes (Figure 7A–C). Similarly, common targets were found in the biological processes of single organism, metabolic, and cellular processes. Common molecular function mRNAs were involved in binding and catalytic activity. Among cellular components, membranes, organelles, and cells were most important (Figure 7D–F). Biological processes of specific (known and novel) miRNA-targeted mRNAs were associated with metabolic processes. Additionally, molecular function mRNAs were involved in transporter, binding, and catalytic activities. Cellular component targets were membrane localized, with commonly expressed miRNAs and cellular mRNAs as targets of specific miRNAs (Figure 7G–I). Overall, all miRNAs observed here targeted mRNAs localized to the cell, organelles, membranes, and macromolecular complexes, with roles in cellular and metabolic processes as well as catalytic, binding, transporter, and molecular transducer activities. At the molecular level, very few targets were related to the nucleic acid-binding transcription factor, molecular function regulator, protein-binding transcription factor, and antioxidant activity terms. At the cellular level, the membrane-enclosed lumen, extracellular region, cell junction, synapse, and virion were represented. In the biological process category, most targets were related to localization, developmental process, multicellular organismal process, response to stimulus, signaling, and cellular component organization or biogenesis (Figure 7G–I).

### 2.5. Stage-Wise Analysis of Highly Expressed mRNAs

To examine the abundance of target gene transcripts based on the stage-wise mRNA FPKM data resulting from high-throughput sequencing, 10 stage-specific highly expressed mRNAs were selected and validated through qRT-PCR. We used *M. incognita* β-actin (BE225475.1) primers as an internal control for normalization of gene expression. Upon experimental verification using qRT-PCR, we observed that all selected stage-wise mRNAs were expressed in *M. incognita*. Of the 10 stage-specific mRNAs, the egg stage included three mRNAs with unknown functions (Minc11396, Minc03738, and Minc10990) (Figure 8A); the J2 stage included four mRNAs with unknown functions (Minc01592, Minc05278, Minc01593, and Minc18141) and two mRNAs with beta-1,4 endoglucanase (Minc19090) and cellulase (Minc09446) functions (Figure 8B); one mRNA in the J3 stage possessed a double-stranded RNA-binding motif (Minc16891) and the other two (Minc00205 and Minc15344) had unknown functions (Figure 8C); one mRNA in the J4 stage had an unknown function (Figure 8D); and among seven mRNAs in the female stage, one mRNA encoded the secreted protein ASP-2 (Minc08697), two mRNAs were uncharacterized proteins (Minc18856 and Minc18984), and the remaining four mRNAs had unknown functions (Minc02270, Minc07322, Minc02526, and Minc02512) (Figure 8E). Overall, the female stage had the highest level of mRNA expression (7), followed by the J2 (6), J3 (3), and egg (3) stages. In the J4 stage, we detected the fewest mRNAs (1) among the stage-specific mRNAs identified in our sequencing data. qRT-PCR results for the selected stage-specific mRNAs are shown in Figure 8. Furthermore, commonly expressed mRNAs were selected and validated; they were expressed at all stages (Appendix A).

### 2.6. Stage-Specific miRNA-Targeted mRNA Expression Analysis

The abundance of target gene transcripts was examined based on miRNA; mRNA results from high-throughput sequencing, represented as read counts, were normalized using the DeSeq2 and FPKM methods. We selected candidates based on inverse expression analysis of the DeSeq2 of upregulated miRNAs and corresponding FPKM of their downregulated target mRNAs. Appendix A is a validated list of miRNA ID, 5′–3′ miRNA sequence, mRNA gene ID, genome contig location, and function for the candidates. We used *M. incognita* β-actin (BE225475.1) primers as an internal control for normalization of gene expression. Upon experimental verification using qRT-PCR, we observed that all selected miRNAs targeting mRNAs were irreversibly expressed at all stages of nematode development. Expression profiles of the 29 miRNAs targeting mRNAs were consistent when expression was compared with the internal standard β-actin (BE225475.1). Upon experimental verification using qRT-PCR, we observed that approximately 60% of the selected stage-wise miRNA-targeted mRNAs were expressed in *M. incognita*. The stage-specific miRNAs targeting mRNAs provided the following results: six miRNA-targeted mRNAs were detected in the egg stage with functions of glucosyltransferase (MIN00371-Minc13648) and patched family protein (MI00171-Minc18132) (Figure 9A). In the J2 stage, the major functions were nematode cuticle collagen (MI03464-Minc01401) and beta-tubulin (MIN00380-Minc13117) (Figure 9B). In the J3 and J4 stages, fucosyltransferase (MI02061-Minc00091), hypothetical protein (MI02049-Minc00288), ubiquitin-conjugating protein (MI03255-Minc00661), and unknown function (MI01808-Minc02324) (Figure 9C,D) were the major functions. In the female stage, among five mRNAs, nematode cuticle collagen (MI01064-Minc12284) and putative secretion protein (MI02712-Minc11857) (Figure 9E) were stage-specific. Our experimental validation showed that few miRNA-targeted mRNAs were expressed during other stages. These mRNAs had their lowest expression levels during the stages in which they were characterized as stage-specific according to high-throughput sequencing data. Selected qRT-PCR results for stage-specific miRNA-targeted mRNAs are shown in Figure 9.

## 3. Discussion

*M. incognita* is an economically important pathogen and one of the most important plant pathogens worldwide. Previous studies regarding the transcriptome of *M. incognita* were limited to juvenile stages J2 and J3 [13,14]. These studies indicated an abundance of cytoskeletal and ligand-binding proteins during the J2 stage, as well as high expression of host immune response genes and parasitic functional glutathione-S-transferases in the J3 stage. In the current study, we conducted RNA sequencing of *M. incognita* in all developmental stages. In general, obligate parasites have compact genomes, and members of the genus Meloidogyne reportedly have the smallest genomes among plant parasites [8,22,23]. Compaction has been proposed as the reason for reduced genome size in organisms that have small genomes with high gene density. The previously reported draft genome studies of nematode revealed 16,419, 18,074, 14,420, 18,348, and 19,212 genes from *M. incognita*, *G. pallida*, *B. xylophilus*, *M. hapla*, and *B. malayi*, respectively [2,8,24,25,26]. Overall, 17,423 transcripts were commonly expressed across all stages of *M. incognita* in our study (Figure 2). As a result, the transcripts obtained in our results are also close to the values in other nematodes. 

Alignment of the sequencing results to available databases indicated that our sequences were orthologous to the sequences of *B. malayi* (3729 hits), *C. elegans* (2489 hits), and *C. briggsae* (1871 hits), comprising 46.43% of transcripts (Appendix A). The filarial nematode *B. malayi* also exhibits a parasitic lifestyle with infective and reproductive stages, which may explain the similarity in expressed genes. Annotation of the mRNA sequences indicated that a large number of mRNA sequences had no matches in the database search (Appendix A). The same phenomenon was observed in previous transcriptional studies of *M. incognita* and other nematodes [8,9,10,11,13]. Elevated expression of actin and the high-mobility group of mRNAs in the egg stage suggest that cell proliferation and DNA-associated processes were active. Annotation of the RNA sequences also indicated the presence of the collagen family protein COL-45 and alpha-(1,3)-fucosyltransferase transcripts; the alpha-(1,3)-fucosyltransferase in *M. incognita* is similar to fut-1 in *C. elegans*, which is expressed in the egg stage. The J2 stage had abundant FMRF-amide-like peptides, which play key roles in neuroregulatory functions. Cellulases similar to the cellulases in *Meloidogyne javanica* (β-1,4-endoglucanase) and *Heterodera glycines* (guanylate cyclase) were also expressed at the J2 stage. During infective stages of the nematode, cellulases can aid the infection of plant tissues, and several cellulases and endoglucanases have been reported in the stylet secretions of the cyst nematode *Globodera rostochiensis* [27,28]. Therefore, motility and infection may be the crucial functional roles of genes expressed at the J2 stage. The J3 and J4 stages exhibited large numbers of collagen and actin-related mRNAs, which are involved in nematode growth and metamorphosis. The cytochrome P450 and cuticle collagen family proteins CYP-23A1 and COL-149, as well as poly(A) polymerases orthologous to the polymerases in *C. briggsae*, were expressed in the J3 stage, whereas several zinc finger proteins and zinc metalloproteinases were specific to the J4 stage. The female-stage nematode also possessed large numbers of cuticle collagen mRNAs, SEC-2-like proteins (which control vesicular transport), and nucleic acid-binding KH domains (Figure 5, Table 4). Differential expression patterns of miRNAs and their effects on mRNA levels showed that several miRNAs were expressed at specific stages of nematode development; these miRNAs may regulate mRNA expression. We observed 1218 known miRNAs and 253 novel miRNAs that targeted mRNAs in *M. incognita*. By comparing the expression levels of miRNAs and mRNAs, we found 76 known and 4 novel miRNAs regulating more than fourfold changes in the expression levels of their target mRNAs.

Among the targeted mRNAs with fourfold changes, the egg and female stages had the greatest number and diversity of targets compared with all other stages. In the egg stage, the greatest numbers of targets were unknown or putative uncharacterized proteins from *C. elegans* and *C. briggsae*. Known targets in the egg stage included carboxypeptidase, a regulator of the nonsense transcripts homolog; β-1,4-N-acetylgalactosaminyltransferase, a metabolite transporter protein homolog; putative esophageal gland cell secretory, helicase-like, and cytochrome P450 family 33-related proteins; and ecdysteroid UDP-glucosyltransferase from *C. briggsae*. In the J2 stage, highly targeted known proteins included β-tubulin, histone-lysine N-methyl transferase, collagen, ubiquitin, and parasitism-related Vap-1 homolog proteins. The J3 and J4 stages had fewer targets that showed significant changes in expression (fourfold). Fucosyltransferase family proteins and guanylate cyclase were major targets in the J3 stage, whereas Nk homeobox protein was the major target in the J4 stage. In the female stage, most target proteins were cuticle- and collagen-related proteins; other targets included the PAN domain, ion channel receptor, HAT family dimerization domain, leucine-rich repeats, RNase L inhibitors, and TRP homologous cation channel protein.

Quantitative analysis of expression using PCR validated that stage-specific miRNAs targeting mRNAs identified in the sequencing-based analysis were expressed in *M. incognita*. We confirmed that these miRNAs targeting mRNAs were authentic; 29 miRNAs targeting mRNAs in the egg, J2, J3, J4, and female stages exhibited stage-specific expression patterns in sequencing reads and corresponding patterns in quantitative PCR analysis. Moreover, selected mRNAs—identified as stage-specific through digital expression analysis based on RNA sequencing—were expressed at all stages. However, the functions of these miRNAs that target mRNAs in *M. incognita* remain unclear. In summary, the present study aimed to supplement existing miRNA data regarding *M. incognita* to identify targeted mRNAs at all stages of development. The data presented here will be useful for future nematologists when selecting potential regulatory networks for targeted control of nematode-induced agricultural losses.

## 4. Materials and Methods

### 4.1. Biological Sample Preparation

*M. incognita* was collected from roots of an infected tomato plant (*Solanum lycopersicum* var. Rutgers) grown in our laboratory greenhouse at a constant 25 °C. The method used to collect each stage of nematode samples was described in our previous study [20]. In brief, eggs were washed from the infected roots and treated with 10% NaClO for 5 min, then excess water was passed through a 25 µm mesh to collect the eggs, which were purified through 35% sucrose gradient centrifugation at room temperature. J2 samples were collected by hatching eggs at 25 °C for 5 days in autoclaved distilled water, and samples were collected using 5–7 layers of Kimtech Science Wipers on a Petri dish placed on a laboratory bench. To collect the J3, J4 and female stages, infected roots were washed, chopped, and treated with 7.7% cellulase and 15.4% pectinase, then washed and filtered through a 75-µm filter. The samples stored on the filters were re-rinsed in water, and nematodes were hand-picked using a pipette under a stereomicroscope.

### 4.2. Stage-Wise RNA Extraction and Sequencing

Nematode samples were ground with a mortar and pestle in liquid nitrogen; an approximately 200 mg sample was used for RNA extraction for each of the five developmental stages (i.e., egg, J2, J3, J4 and female) with three biological replicates collected. Total RNA from the samples was quantified using a Nanodrop spectrophotometer (Thermo Scientific, CA, USA); quality was assessed with the RNA 6000 Nano Kit (Agilent Technolgies, Santa Clara, CA, USA) and Bioanalyzer 2100 (Agilent Technolgies, Santa Clara, CA, USA). One microgram of treated RNA was used for next-generation sequencing; libraries were generated using the TruSeq RNA Sample Prep Kit (Illumina Inc., San Diego, CA, USA), in accordance with the manufacturer’s instructions. In brief, poly(A)-containing RNA molecules were extracted using a poly-T oligonucleotide attached to magnetic beads. The purified total poly(A) RNA was fragmented into small pieces using divalent cations at elevated temperature. The cleaved mRNA fragments from each stage were treated with DNase I at 37 °C for 30 min to remove residual DNA. Using random primers, the first-strand cDNA was synthesized and the fragments were purified with the QIAquick PCR extraction kit. Finally, elution buffer was added for end repair and addition of poly(A). Subsequently, the repaired short fragments were attached to sequencing adapters, and then, each library was assigned an adjoining distinct multiplex identifier tag. The resulting cDNA libraries were sequenced with the Illumina HiSeq™ 2500 system.

### 4.3. Assembly and Genome Mapping

Complete paired-end sequences were obtained as individual fasta files (forward and reverse) from the images using CASAVA v.1.8.2 base-calling software with an ASCII Q-score offset of 33. Raw reads sequences were trimmed by the parameters, quality trimming based on Phres quality scores (Q ≤ 20), adaptor trimming and minimum length discard (<90 bp) using the CLC genome cell (v. 4.0). To characterize the quantitative expression patterns of individual genes, the clean sequence reads from five libraries with three replicates (egg, J2, J3, J4 and female) were mapped individually to the reference genome under the guidance of gff3 using Bowtie from TopHat (v. 2.0.8) with the following parameters: mate-inner-dist (100), mate-std-dev (200), splice-mismatch (1), library-type (fr-unstranded), and microexon-search. Differential expression between pairs was calculated using Cuffdiff (v. 2.1.1) [29]. Final sequences were filtered with the following criteria: FPKM ≥ 0.3 [30], FDR (q value) < 0.01 and log_2_ (FC) ≥ 1.00.

### 4.4. Functional Annotation and Expression Analysis

Each gene sequence was subjected to functional annotation using Pedant-Pro suite (Biomax Informatics AG, Martinsried, Germany) [31] with customized database parameters: eukaryote non-plant database; analysis type: Intronic rich genes (nematodes, *C.*
*elegans*); genetic code: stranded, blastp, and Blast2GO [32] and other default parameters. Functional annotations such as GO terms and Clusters of Orthologous Groups were assigned to the gene sequences using Pedant-Pro default scores. The expression patterns of genes across different stages were compared using a heatmap generated with MeV software (v. 4.8.1).

### 4.5. miRNA Sequencing and Target Prediction

miRNA sequencing and classification at various stages of the *M. incognita* lifecycle were described in detail in our previous report [3]. In total, 3107 miRNAs were used for target prediction. To obtain the 3′ UTR region of each gene, prediction was conducted using Python scripts as follows. Initially, gene coordinates were taken from the GFF3 file for *M. incognita* available from WormBase (ASM18041v1a) [2], and the 3′ flanking region from the stop codon to a maximum of 5000 bases downstream was searched for 16 polyadenylated site motifs. The region from the stop codon to the first polyadenylated site motif plus 20 downstream bases was considered the 3′ UTR for each gene. The polyadenylated site motifs were taken from the *C. elegans* genome [33]. 

To predict potential targets, we used bioinformatics software including miRanda, RNAhybrid, and PITA. First, we used two different methods, miRanda [34] and RNAhybrid [35,36], to predict miRNA targets by detecting the most energetically favorable hybridization sites for small RNA within the 3′ UTR based on sequence similarity features [37]. The default parameters for miRanda were used and an output filter was applied (ΔG ≥ −25.0 kcal/mol). The parameters for RNAhybrid were number of hits per target = 1, energy cutoff ≥ −25.0 kcal/mol, and maximal internal or bulge loop size per side = 4. Finally, the common mRNA targets obtained from miRanda and RNAhybrid were used for prediction with PITA, where ΔΔG values below −10 were considered representative of endogenous miRNA expression levels.

### 4.6. Comparison of miRNA and Complementary mRNA Expression

miRNA and mRNA expression patterns were integrated with targets through inverse expression analysis (miRNA upregulated and corresponding target mRNA downregulated) using Python scripts. miRNAs were normalized to obtain DESeq2 and mRNA expression was normalized to obtain FPKM. The miRNA and target expression levels were then compared.

### 4.7. Stage-Wise Gene Expression Analysis

qRT-PCR was performed for 29 miRNAs targeting mRNAs and 20 stage-specific mRNA genes selected based on stage specificity, high expression levels determined using DESeq2, and mRNA expression levels determined using FPKM values. qRT-PCR was performed using the CFX96 Real-Time PCR detection System (Bio-Rad Laboratories, Hercules, CA, USA) with SYBR Premix (Toyobo, Osaka, Japan). The qRT-PCR mixture contained 5 nmol of each primer, as well as SYBR Premix. The reactions were run with the following cycling conditions: denaturation at 95 °C for 5 min, followed by 45 cycles of denaturation at 95 °C for 15 s and annealing at 60 °C for 30 s. The amplification products of stage-specific miRNAs targeting mRNAs were used to create qRT-PCR primers. Primer pairs were designed using the PrimerQuest Tool (Integrated DNA Technologies, Coralville, IA, USA); all primers used in the study are listed in Appendix A. Validated *M. incognita* β-actin (BE225475.1) primers in our previous study [38,39] were used as an internal control for normalization of gene expression. The 2^−ΔΔCt^ method of Livak and Schmittgen (2001) [40] was used to quantify relative changes in gene expression levels.

## Figures and Tables

**Figure 1 ijms-22-07442-f001:**
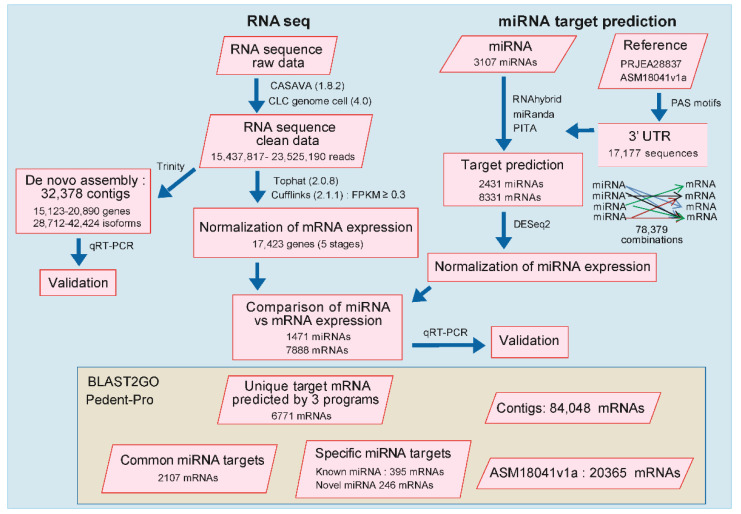
Flowchart representing the overall process of the current study. In silico analysis of miRNA-targeted mRNA prediction from the reference genome (ASM18041V1a) [2]; further combinations of miRNA-targeted mRNAs were elucidated. Illumina sequence data were used as the raw data; these data were cleaned using multiple software programs (RNAhybrid, miRanda, and PITA) and differentially expressed mRNAs were analyzed with a Venn diagram and heatmap. Expression levels of miRNA and mRNA based on sequencing results were normalized using DESeq2 and the fragments per kilobase of transcript per million mapped reads (FPKM) methods, respectively. Further miRNA-targeted mRNAs were validated through qRT-PCR for all developmental stages of *M. incognita*. Sequencing results were annotated with BLAST2GO using data for specific, common, and unique miRNA-targeted mRNAs.

**Figure 2 ijms-22-07442-f002:**
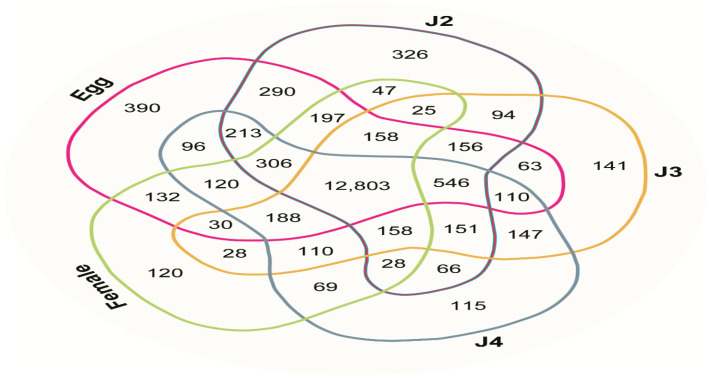
Venn diagram illustrating the distribution of reads across the developmental stages of *M. incognita* (egg, J2, J3, J4, and female), indicating commonly expressed and stage-specific mRNAs (FPKM ≥ 0.3).

**Figure 3 ijms-22-07442-f003:**
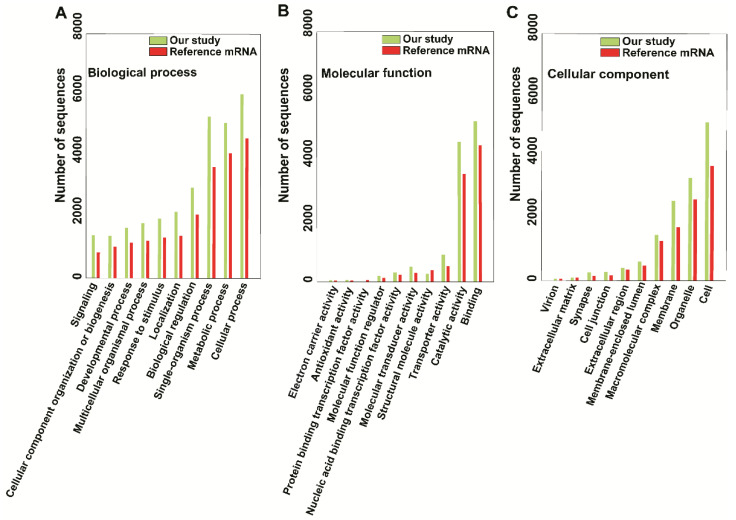
Comparison of top Gene Ontology (GO) mapping results. The results are summarized into three main categories: (**A**) biological process, (**B**) molecular function, and (**C**) cellular component. Dark green color indicates the number of sequences used in our study; dark red color indicates the number of sequences in the reference study (ASM18041V1a).

**Figure 4 ijms-22-07442-f004:**
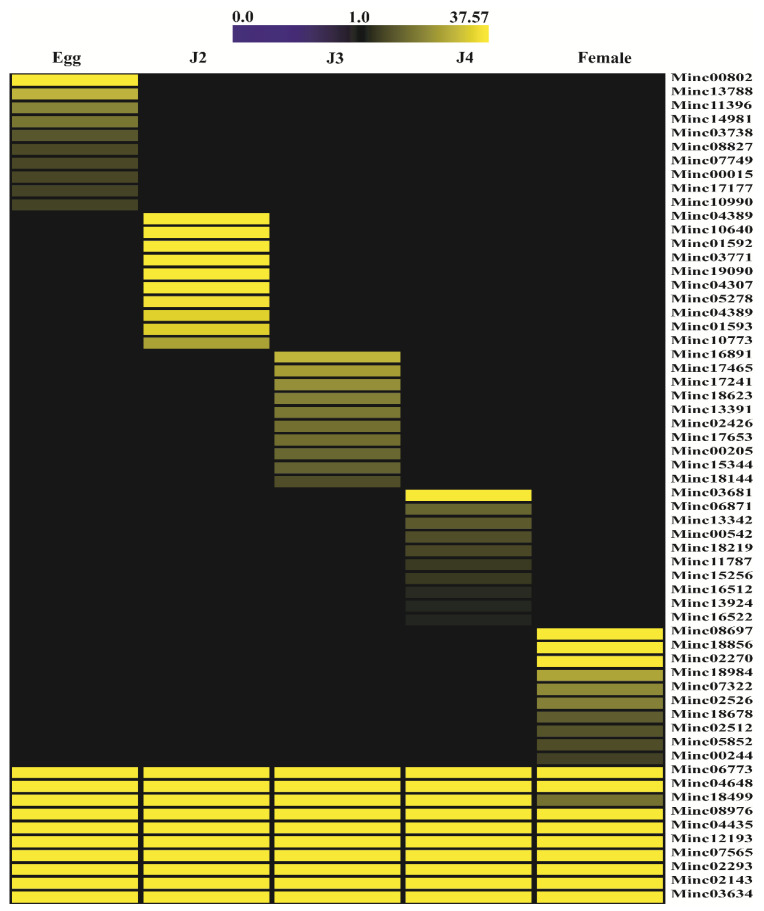
Heat map of selected stage-specific highly expressed mRNA (FPKM) profiles at all developmental stages. “Minc” indicates a stage-specific transcript gene ID. Darker yellow colors indicate higher expression levels of genes, and black indicates no expression. The selected mRNAs were validated using quantitative real-time PCR (Figure 8).

**Figure 5 ijms-22-07442-f005:**
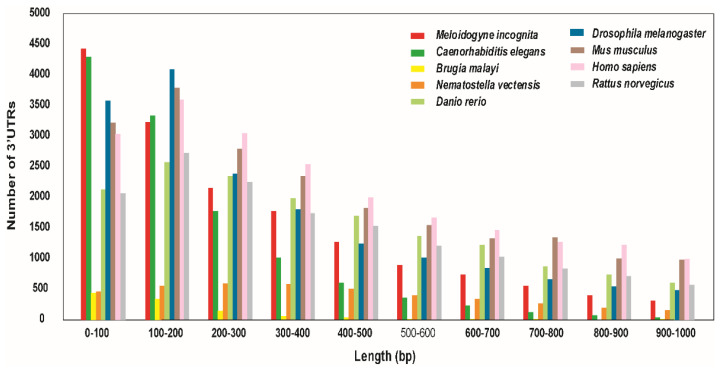
Comparison of 3′ UTR sequence length distributions among nematode species (*C. elegans*, *M. incognita*, and *B. malayi*), mammalian groups (human, mouse, and rat) and others (starlet sea anemone, fruit fly, and zebrafish). Overall, *M. incognita* had longer 3′ UTR sequences than *C. elegans*, except for sequences 100–200 bp in length.

**Figure 6 ijms-22-07442-f006:**
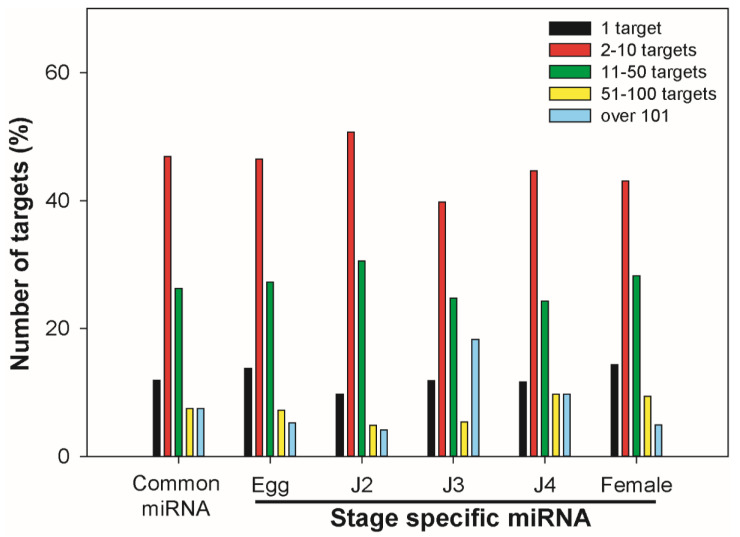
Percentage chart showing the number of mRNA targets of miRNAs expressed in *M. incognita*. The number of targets for each miRNA was determined at all stages, and the most miRNAs were found to target 2–10 mRNA sequences. The second highest percentage was for miRNAs targeting 11–50 mRNA. These results indicated high target range diversity among miRNAs.

**Figure 7 ijms-22-07442-f007:**
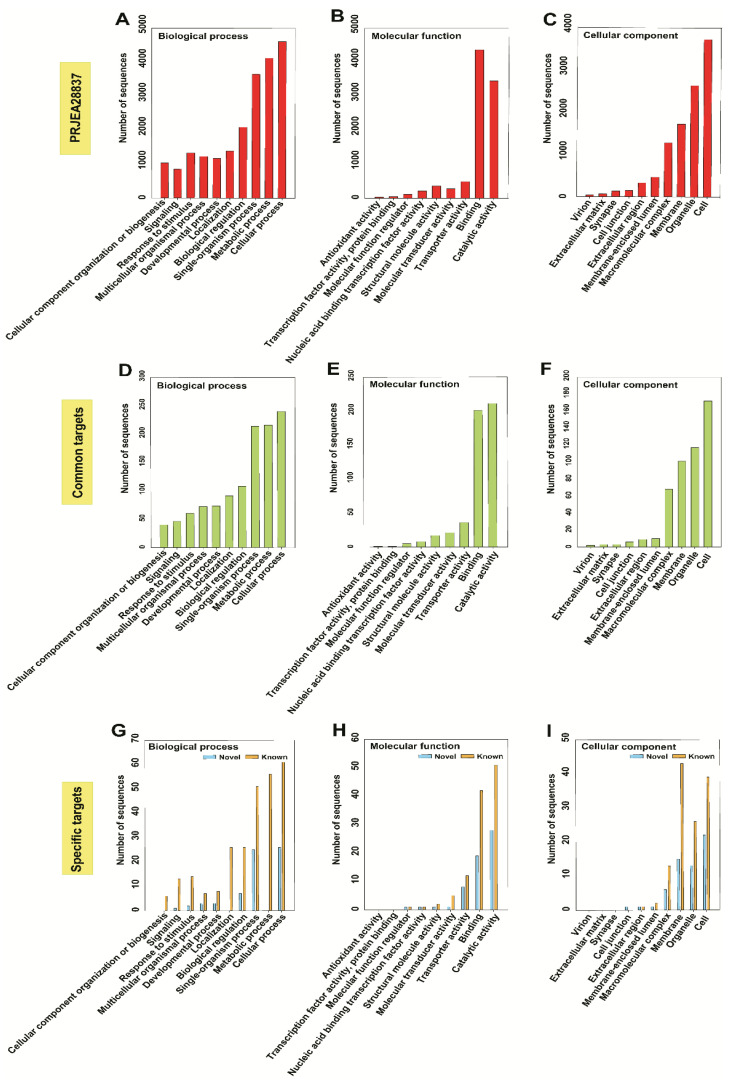
GO distribution of miRNA-targeted mRNAs predicted using miRanda, PITA, and RNAhybrid programs. (**A**–**C**) reference targets; (**D**–**F**) miRNA targets common across all stages; (**G**–**I**) known and novel miRNA targets specific to stages categorized into biological processes, molecular functions, and cellular components.

**Figure 8 ijms-22-07442-f008:**
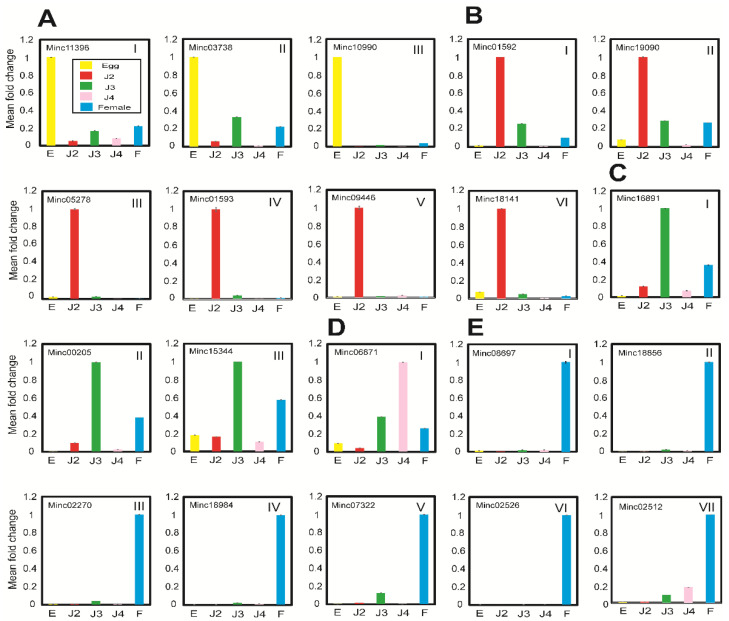
Real-time expression of stage-specific highly expressed mRNAs. (**A**) egg stage; (**B**) J2 stage; (**C**) J3 stage; (**D**) J4 stage; (**E**) female stage. Bar color indicates developmental stage: yellow, egg stage; red, J2 stage; green, J3 stage; pink, J4 stage; and blue, female stage. E: Egg; F: Female.

**Figure 9 ijms-22-07442-f009:**
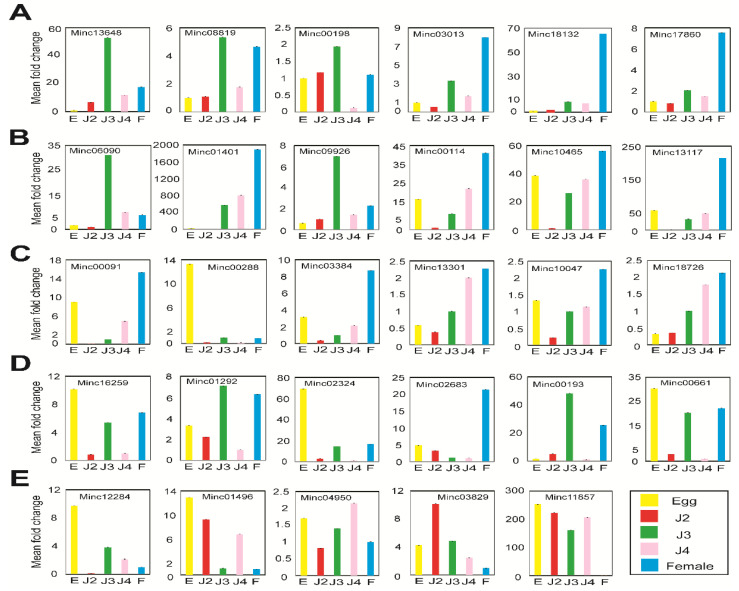
Validation of sequencing and real-time expression of stage-specific miRNAs targeting mRNAs. (**A**) egg stage; (**B**) J2 stage; (**C**) J3 stage; (**D**) J4 stage; (**E**) female stage. Bar color indicates developmental stage: yellow, egg; red, J2; green, J3; pink, J4; and blue, female. E: Egg; F: Female.

**Table 1 ijms-22-07442-t001:** Stage-wise statistics of *M. incognita* mRNA.

Stage	Raw Seq *	High-Quality Seq *	Mapped Reads	Paired Reads	Mapping Ratio
Egg	70,310,943	56,902,920	35,233,964	23,525,190	61.92%
J2	61,150,827	50,762,456	26,034,163	15,437,817	51.31%
J3	51,342,127	40,968,532	23,321,460	15,504,737	56.98%
J4	58,016,948	47,309,223	26,676,718	15,897,997	56.40%
Female	64,618,511	51,730,234	28,551,685	16,472,789	55.23%

* Obtained from our previous report, Choi et al. 2017 [20].

**Table 2 ijms-22-07442-t002:** Stage-wise distribution of *M. incognita* mRNA gene expression. The number of genes expressed at each stage is based on the threshold of FPKM ≥ 0.3.

Stage	Egg	J2	J3	J4	Female	Unique
Number of genes	15,798	15,564	14,908	15,226	14,519	17,423

**Table 3 ijms-22-07442-t003:** Highly expressed mRNAs (top 5) from each developmental stage (egg, J2, J3, J4 and female).

Transcript ID	Gene ID	Locus *	Hit Description	*E*-Value
Egg				
Minc04648a	Minc04648	MiV1ctg109:43326-44000	Putative uncharacterized protein hmg-1.1 (High mobility group protein 1.1)-*Caenorhabditis elegans*	4.5 × 10^−15^
Minc18499a	Minc18499	MiV1ctg2009:3204-5374	CBR-SQT-2 protein-*Caenorhabditis briggsae*	1.4 × 10^−38^
Minc08976	Minc08976	MiV1ctg306:49411-50078	Acidic ribosomal protein-*Ceratitis capitata* (Mediterranean fruit fly)	1.1 × 10^−18^
Minc06773a	Minc06773	MiV1ctg191:65479-67430	Actin-*Panagrellus redivivus*	0
Minc00328a	Minc00328	MiV1ctg4:84197-88448	NA	NA
J2				
Minc06998a	Minc06998	MiV1ctg201:70539-71397	FMRF-amide-like peptide 14-*Meloidogyne incognita*	1.2 × 10^−52^
Minc18702	Minc18702	MiV1ctg2199:2335-4036	NA	NA
Minc16402	Minc16402	MiV1ctg1163:4158-4357	NA	NA
Minc07370	Minc07370	MiV1ctg219:88182-88607	NA	NA
Minc03882	Minc03882	MiV1ctg84:123564-124307	NA	NA
J3				
Minc18145a	Minc18145	MiV1ctg1750:2047-6839	Nematode cuticle collagen N-terminal domain-containing protein-*Brugia malayi* (Filarial nematode worm)	7.3 × 10^−34^
Minc04518a	Minc04518	MiV1ctg105:26001-27747	Cuticle collagen-*Meloidogyne incognita*	4.5 × 10^−62^
Minc06773a	Minc06773	MiV1ctg191:65479-67430	Actin-*Panagrellus redivivus*	0
Minc18882	Minc18882	MiV1ctg2407:4325-4977	NA	NA
Minc18693	Minc18693	MiV1ctg2190:124-789	NA	NA
J4				
Minc04518a	Minc04518	MiV1ctg105:26001-27747	Cuticle collagen-*Meloidogyne incognita*	4.5 × 10^−62^
Minc01401a	Minc01401	MiV1ctg21:214717-219350	COL-1-*Meloidogyne incognita*	5.9 × 10^−51^
Minc08754	Minc08754	MiV1ctg293:33744-34125	NA	NA
Minc00801	Minc00801	MiV1ctg11:126856-128485	NA	NA
Minc18693	Minc18693	MiV1ctg2190:124-789	NA	NA
Female				
Minc16604	Minc16604	MiV1ctg1209:9257-9418	NA	NA
Minc19059	Minc19059	MiV1ctg2663:360-638	NA	NA
Minc17994	Minc17994	MiV1ctg1660:2568-3249	NA	NA
Minc19141	Minc19141	MiV1ctg2796:3302-3931	NA	NA
Minc18070	Minc18070	MiV1ctg1703:2398-2962	NA	NA

* Reference genome Project ID: PRJEA28837; NA: Function not validated.

**Table 4 ijms-22-07442-t004:** Most abundant shared mRNA contigs among 12,803 commonly expressed mRNAs in all stages of *M. incognita*.

Transcript ID	Gene ID	Locus *	Hit Description	*E*-Value
Minc01401a	Minc01401	MiV1ctg21:214717-219350	COL-1-*Meloidogyne incognita*	5.9 × 10^−51^
Minc04648a	Minc04648	MiV1ctg109:43326-44000	Putative uncharacterized protein hmg-1.1 (High mobility group protein 1.1)-*Caenorhabditis elegans*	4.5 × 10^−15^
Minc01079	Minc01079	MiV1ctg16:17706-18709	SEC-2 protein-*Globodera pallida*	1.6 × 10^−45^
Minc08986	Minc08986	MiV1ctg307:10895-11969	SEC-2 protein-*Globodera pallida*	1.6 × 10^−45^
Minc18499a	Minc18499	MiV1ctg2009:3204-5374	CBR-SQT-2 protein-*Caenorhabditis briggsae*	1.4 × 10^−38^
Minc08976	Minc08976	MiV1ctg306:49411-50078	Acidic ribosomal protein-*Ceratitis capitata* (Mediterranean fruit fly)	1.1 × 10^−18^
Minc16667a	Minc16667	MiV1ctg1222:13554-14957	Vasa-related protein CnVAS1-*Hydra magnipapillata*	1.1 × 10^−19^
Minc06773a	Minc06773	MiV1ctg191:65479-67430	Actin-*Panagrellus redivivus*	0
Minc00801	Minc00801	MiV1ctg11:126856-128485	NA	NA
Minc02143	Minc02143	MiV1ctg39:10470-10990	NA	NA
Minc02293	Minc02293	MiV1ctg42:40801-41759	NA	NA
Minc03882	Minc03882	MiV1ctg84:123564-124307	NA	NA
Minc11652	Minc11652	MiV1ctg495:37571-37982	NA	NA
Minc18693	Minc18693	MiV1ctg2190:124-789	NA	NA

* Reference genome Project ID: PRJEA28837; NA: Function not validated.

**Table 5 ijms-22-07442-t005:** Stage-specific targets of miRNA obtained using miRanda, RNAhybrid and PITA. Prediction of miRNA targets was conducted only for miRNA sequences reported in our pervious study [3] with read count > 10.

Stage	miRNA	miRBase Reference	Sequence	mRNA Gene ID	mRNA Transcript ID	Number of mRNA Hits in Target Prediction	Hit Description
Egg-specific	MI00282, MI00853, MI02967, MIN00081	ptr-miR-3138, hsa-miR-5196-5p, bmo-miR-3374-5p, Novel	AAGGAGGGAGAGGGAATG, AGGGAAGGGAGAGAGGGAGGGG, TGAAGAGCACGGATGTTGAAGGGC, AGGGGAAAGGGCGAGGAGGGG	Minc04914	Minc04914a, Minc04914b, Minc04914c	12	Transcription factor unc-3 (Uncoordinated protein 3) (CEO/E)-*Caenorhabditis elegans*
Egg-specific	MI00870, MI01912, MI02333, MI03713, MIN00025, MIN00050, MIN00126, MIN00199, MIN00335	hsa-miR-5001-5p, hsa-miR-761, mmu-miR-6944-5p, aga-miR-210, Novel, Novel, Novel Novel Novel	AGGGCTGGACTCAACTGCGGATTGCGT, GCAGAGAGGGTGAAACTGAAC, GTGGAGGGGGGGGAGGGCAA, TTGTGGGGTGTCAACGGCATA, AAGGGAAGGGAAGGGAAGGG, ACTAGAGATCGAGCTGGGCCTG, CCGGACTGGATCCGGCCGGATTT, GGGGGATGACATTGTAATTGAACA, TGGTGTATCTTGTATTGTGGGTA	Minc08454, Minc00238	Minc08454, Minc00238	9	CBR-HSP-12.2 protein-*Caenorhabditis briggsae*
Egg-specific	MI00353, MI00820, MI01640, MI02876, MI03247, MI03634, MIN00334	cte-miR-2f, cbr-miR-73b, cgr-miR-671-5p, mmu-miR-6336, gga-miR-6645-5p, hsa-miR-3064-3p, Novel	AATCAAGTCGGATTTGGTTGAT, AGGCAAGATGTTGGCATTGTCGAT, GAAAGCCTGGAGCGGCTGGAGTG, TCTCGGATTTAGTAAGAACGGCC, TGGAGGATGTAGCAGTGGTGGCGGA, TTGCCCAACTGAACAATCCTTACA, TGGTCTGTCAGTCATAGGTTAT	Minc13215, Minc14656	Minc13215, Minc14656	7	CAP-Gly domain-containing protein-*Brugia malayi* (Filarial nematode worm)
Egg-specific	MI00171, MI02031, MI02967	ame-miR-6053, hsa-miR-1273f, bmo-miR-3374-5p	AACGAAGACCGCGGCGGAGCTGT, GGAGACGGATGGTTGCAG, TGAAGAGCACGGATGTTGAAGGGC	Minc05892, Minc03605	Minc05892, Minc03605	6	DNA excision repair protein ERCC-6, putative-*Brugia malayi* (Filarial nematode worm)
Egg-specific	MI01244, MI01640, MI01891, MI02222, MI02986, MIN00058	gga-miR-1583, cgr-miR-671-5p, mml-miR-7175-3p, dme-miR-4971-5p, ptr-miR-4660, Novel	CAAGGGACTGGGACGGCA, GAAAGCCTGGAGCGGCTGGAGTG, GCAACATATGGTTGAGAGGACTGG, GGTTGAGTTGTCGGAGGTGGCGGA, TGACAGATTGGTGGAAAGATTGGA, AGAGACCGGACTGGGAGTGTCG	Minc07882	Minc07882	6	Mothers against dpp protein-*Aedes aegypti* (Yellow fever mosquito)
J2-specific	MI02742	hsa-miR-8063	TCAAAAAGAAGTCGGGGGTT	Minc15286	Minc15286a, Minc15286b, Minc15286c	3	SAC3/GANP family protein-*Brugia malayi* (Filarial nematode worm)
J2-specific	MI00606	ggo-miR-320b	AGAAGTTGGATTGAGAGGGA	Minc02238, Minc14931	Minc02238, Minc14931	2	Ab1-108-*Rattus norvegicus* (Rat)
J2-specific	MI02073, MI03005	gga-miR-6627-3p, ppy-miR-4667	GGATGAAGACGATGATGCTGAGGA, TGACTGGAGGAGGAGGAGGAA	Minc19110	Minc19110	2	CG32584-PB, putative-*Brugia malayi* (Filarial nematode worm)
J2-specific	MI00699	pma-miR-4546	AGCAGAAGTCGTAGTGGAAG	Minc04635, Minc03649	Minc04635, Minc03649	2	CG6766-PA, putative-*Brugia malayi*
J2-specific	MI02028	bta-miR-2486-5p	GGAGAAGACGGGGGTGGTGGTGGG	Minc11989	Minc11989a, Minc11989b	2	Dnaja2-prov protein-*Xenopus laevis* (African clawed frog)
J3-specific	MI01281, MI03334	gga-miR-6711-5p, ame-miR-3739	CAGATTGGGATAGAAAGAAGCA, TGGGAGGGGGGAGAGAGAT	Minc11006	Minc11006	2	CBR-WWP-1 protein-*Caenorhabditis briggsae*
J3-specific	MI02003	dsi-miR-1003-5p	GCTGGGCTGTCTGGTGTGGTTGG	Minc06126, Minc02988	Minc06126, Minc02988	2	MBOAT family protein-*Brugia malayi* (Filarial nematode worm)
J3-specific	MI03334, MI03358	ame-miR-3739, hsa-miR-6887-5p	TGGGAGGGGGGAGAGAGAT, TGGGGAGGAAGAGGAGGAGGACA	Minc10383	Minc10383	2	MSP domain protein-*Brugia malayi*
J3-specific	MI02164, MI02411	hsa-miR-6752-5p, gga-miR-6607-5p	GGGGGTTGTGGAGGAGGGGG, GTTGGTGGAGGAAGTGGAGGTGG	Minc16144	Minc16144	2	Putative uncharacterized protein ptr-14-*Caenorhabditis elegans*
J3-specific	MI02039, MI03235	mmu-miR-6370, mmu-miR-1940	GGAGGAACGAGCAAAGGAGGAACGC, TGGAGGACGAGGAGGTGGAGGGTT	Minc04418	Minc04418	2	Ras-related protein Rab-35, putative-*Brugia malayi* (Filarial nematode worm)
J4-specific	MI03383	dme-miR-1003-5p	TGGGGGTTGGTGTGGTTGG	Minc10127	Minc10127a, Minc10127d, Minc10127e	3	Troponin t protein 2, isoform a-*Caenorhabditis elegans*
J4-specific	MI01803	sme-miR-13-5p	GAGGCTTTGGAGGCGGCTGTGATACT	Minc11517, Minc15705	Minc11517, Minc15705	2	RNA polymerase Rpb7, N-terminal domain-containing protein-*Brugia malayi* (Filarial nematode worm)
J4-specific	MI00298	mmu-miR-6978-5p	AAGGGATGAAGGGAGAAGCTGGT	Minc09642	Minc09642	1	40S ribosomal protein S2, putative-*Brugia malayi*
J4-specific	MI00936	bmo-miR-750-5p	AGTGGACAGGGAGATGCTTGGAAA	Minc08802	Minc08802	1	Alpha amylase, catalytic domain-containing protein-*Brugia malayi*
J4-specific	MI03777	crm-miR-239b	TTTGTACTAGCCAAAATCTCTGCA	Minc14488	Minc14488	1	Carbonic anhydrase like protein 2, putative-*Brugia malayi* (Filarial nematode worm)
Female-specific	MI03351	hsa-miR-4467	TGGGCGGCGGTAGGTTGGGCT	Minc06509	Minc06509a, Minc06509d	2	Probable molybdopterin-binding domain-containing protein-*Brugia malayi* (Filarial nematode worm)
Female-specific	MI02846, MIN00138	hsa-miR-6749-5p, Novel	TCGGGTGGGGGTGGGGGAGGC, CGGGGAGCGTTGGAGGATGACT	Minc15474	Minc15474	2	SUMO-activating enzyme subunit 2 (EC 6.3.2.-) (Ubiquitin-like 1 activating enzyme E1B) (Anthracycline-associated resistance ARX)-*Homo sapiens* (Human)
Female-specific	MIN00243, MIN00364	Novel Novel	GTTGATTCTCGGTCTCGGTCTC, TTGGGTGCTGGTGGAGGAGGGG	Minc09957	Minc09957	2	Transient-receptor-potential-like protein (TRP homologous cation channel protein 1)-*Caenorhabditis elegans*
Female-specific	MIN00243	Novel	GTTGATTCTCGGTCTCGGTCTC	Minc09158	Minc09158	1	1,4-alpha-glucan branching enzyme, putative (EC 2.4.1.18)-*Brugia malayi* (Filarial nematode worm)
Female-specific	MIN00207	Novel	GGTGTCGATGTAGCTAGTGGTGAC	Minc00048	Minc00048	1	Adenylyl cyclase protein 2-*Caenorhabditis elegans*

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
