# Peer review of "Transcriptome Analysis and miRNA Target Profiling at Various Stages of Root-Knot Nematode Meloidogyne incognita Development for Identification of Potential Regulatory Networks"

_ijms, 2021, doi:10.3390/ijms22147442_

Round 1
Reviewer 1 Report
This is an interesting topic: as the authors note, and I think that the manuscript will provides new insights into the molecular mechanisms occurring at various developmental stages of the RKN life cycle. And I enjoyed what clearly has the potential to be a very rich data set. I believe that the manuscript could be published prior to some revision
I have some reservations about the way the results are presented. Your findings represent the story you are going to tell in response to the research questions you have answered. Thus, you sholuld want to organize that story in a way that makes sense to you and will make sense to your reader. Thus giving justifications (i.e. why you have conducted each experiment) in each part of the results will be o great help.
Author Response
First of all, we would like to thank the editor and reviewers for their efforts and suggestions on our manuscript, which have certainly helped to improve the quality of the manuscript. On the basis of the reviewer’s comments, we have revised our manuscript to improve its current quality. Changes / justifications to their comments and suggestions are provided below.
Reviewer 1:
1. I have some reservations about the way the results are presented. Your findings represent the story you are going to tell in response to the research questions you have answered. Thus, you sholuld want to organize that story in a way that makes sense to you and will make sense to your reader. Thus giving justifications (i.e. why you have conducted each experiment) in each part of the results will be o great help.
Author reply: We would like to thank the reviewer for the suggestion to further improvement of our manuscript. As per reviewer suggestion we have updated the purpose of the experiment in the each section of results part.
Reviewer 2 Report
In this article submitted to International Journal of Molecular Sciences, Vimalraj Mani and colleagues report a transcriptomic analysis, including miRNAs and their targets, during different developmental stages of the root-knot nematode Meloidogyne incognita.
My main criticism comes from the sampling itself. Indeed, I wonder to what extent the methods used for sampling can affect (as a stress) the transcriptome of the different stages and how the authors can decipher it from the normal development of M. incognita. It seems to me very important to clarify this aspect in particular because different protocols are used for the different stages. Otherwise, the study was well designed. I have noted below shortcomings that the authors should consider to improve their manuscript.
Introduction
-L48: the authors should provide a reference about the definition of the different developmental stages.
Results
-L107, In Figure 1 (also in L223 and L507 in MM): the authors mention “novel” miRNAs. It is ambiguous, since it could suggest that novel miRNAs were identified in the present study, whereas they were identified in their 2016 paper. I consider that these miRNAs are no longer novel.
-L113: It is written that expression levels for miRNAs and mRNAs were normalized using FPKM and DESeq2 methods, respectively, whereas it is written the contrary in the MM section.
Furthermore, I am not sure that this kind of information should be included in the results section.
-L132-135: The sentence is a repetition of the MM section.
-L146-147: The sentence ending with “’(390)” is not clear. I guess that the authors speak about specific transcripts.
-L163: The mRNA reference sequences information given in L175 should be better given here.
-L190. Proportions instead of numbers will be more useful to compare the two categories.
-L232-234: The sentence is not clear to me. Do the authors mean that the 3’ UTR regions were predicted for 20,365 genes? Also, I don’t understand the second part of the sentence.
-L237, Figure 5: The label of the Y axis “total number of frequency” is not clear to me.
-L242 the sentence is partly a repetition of those in line 223-224 and 224-225.
-L275, Figure 6: The Y-axis label is missing an "s" at the end of "target".
-L312. The sentence starting by “To normalize…” is a technical information that should not be included in the results section. Ditto for L344.
Discussion
-L382-384: I don’t understand the sentence and how the authors can conclude it supports the hypothesis that the genome of M. incognita is compact. Do the authors mean that the majority of the genes were expressed in all stages and only a very few were stage-specific, meaning that the transcriptome varies little between the different stages with a reduced repertoire of expressed genes? The authors should develop their idea and better explain this part.
Materials and Methods
-L487: what is the “bad fraction”? Sequences with a low mean quality?
-L488, I don’t understand what the authors mean by “low-quality bases…were removed”. Do they mean that low-quality bases are replaced by undefined nucleotides (N)? Or do they cut the 3’ extremity of reads when the quality is below a phred score of 20?
-L491: the authors should provide information related to the reference genome they used.
-L528. The sentence is not clear. What does the authors mean by “DESeq2 values”? Normalized abundances of reads? P-values?
-L544: there is no information about the PCR efficiency of each primer pair, whereas obtaining an efficiency close to 2 is a prerequisite to apply the 2-DDCq method. They could calculate the efficiency from data obtained with different template dilutions using the formula (E) = 10−1/slope as described by Pfaffl (2001) [1]. The authors should also provide stability values to justify their choice of the reference gene (p14, l3), using software such as geNorm or NormFinder.
- Pfaffl MW. A new mathematical model for relative quantification in real-time RT-PCR. Nucleic Acids Res. 2001;29:e45.
Author Response
First of all, we would like to thank the editor and reviewers for their efforts and suggestions on our manuscript, which have certainly helped to improve the quality of the manuscript. On the basis of the reviewer’s comments, we have revised our manuscript to improve its current quality. Changes / justifications to their comments and suggestions are provided below.
Reviewer 2:
1. My main criticism comes from the sampling itself. Indeed, I wonder to what extent the methods used for sampling can affect (as a stress) the transcriptome of the different stages and how the authors can decipher it from the normal development of M. incognita. It seems to me very important to clarify this aspect in particular because different protocols are used for the different stages. Otherwise, the study was well designed. I have noted below shortcomings that the authors should consider to improve their manuscript.
Author reply: Thank you very much for the comments. First of all, life of cycle plant parasitic nematode (M. incognita) contains five different developmental stages which include (Egg, J2, J3, J4 and Female) (Reference paper: Abad, P et al 2008, Genome sequence of the metazoan plant-parasitic nematode Meloidogyne incognita). Each stage of samples were collected using different methods, because collection samples required huge number of samples. Initially we have monitored stage-wise sample through microscope. In case of egg stage it was developed in the outside of root (i.e Female sac also called as eggmass), while in J2 stage were collected by hatching the eggs at 27°C for 5-7 days in double distilled autoclaved water in the rotor incubator with 1000 RPM and filtering using 5?7 KIMTECH ScienceWipers on a Petri dish placed on a lab table. Around 2 and 6 weeks after infection presence of J3, J4, and female stage nematodes was identified by manual inspection of root-knots under a stereo microscope. Nematodes at J3, J4, and female were collected using the following protocol. The harvested roots containing root-knots were washed, chopped and treated with 7.7% cellulase and 15.4% pectinase followed by rinsing in running water and filtering through a 75μM mesh. For every five roots, 15mL of cellulase and 30mL of pectinase was used during enzyme treatment. Rinsing of the roots following enzyme treatment was carried out over a filter (75μMmesh) and nematodes retained on the surface of the filter were suspended in distilled water and handpicked using a pipette under a stereo microscope. In our earlier studies (Choi et al 2017, RNA-seq of plant-parasitic nematode Meloidogyne incognita at various stages of its development) contain detailed protocol and also cited in current manuscript.
2. L48: the authors should provide a reference about the definition of the different developmental stages.
Author reply: Thank you for the comments. As per reviewer suggestion we incited the reference for developmental stages of nematode.
3. L107, In Figure 1 (also in L223 and L507 in MM): the authors mention “novel” miRNAs. It is ambiguous, since it could suggest that novel miRNAs were identified in the present study, whereas they were identified in their 2016 paper. I consider that these miRNAs are no longer novel.
Author reply: Thank you for the comments. We removed the known and novel and changed to identified miRNA in throughout manuscript. Also we changed Figure 1 2724, novel and 383 known miRNA to 3107 miRNA.
4. L113: It is written that expression levels for miRNAs and mRNAs were normalized using FPKM and DESeq2 methods, respectively, whereas it is written the contrary in the MM section.
Author reply: Thank you for your suggestion. We have changed as per like in materials and method.
5. L132-135: The sentence is a repetition of the MM section.
Author reply: Thank you for the comments. We have removed the repeated sentence in the Table.1 legends.
6. L146-147: The sentence ending with “’(390)” is not clear. I guess that the authors speak about specific transcripts.
Author reply: Thank you so much for your comments. Yes, it is specific transcript of egg stage. Now the sentence was rewritten.
7. L163: The mRNA reference sequences information given in L175 should be better given here.
Author reply: Thank you so much for your comments. We put reference in line 163.
8. L190. Proportions instead of numbers will be more useful to compare the two categories.
Author reply: Thank you so much for your comments. We have put the GO values of references in the sentences.
9. L232-234: The sentence is not clear to me. Do the authors mean that the 3’ UTR regions were predicted for 20,365 genes? Also, I don’t understand the second part of the sentence.
Author reply: Thank you for your valuable comments. We rewrite the sentence and demonstrated well.
10. L237, Figure 5: The label of the Y axis “total number of frequency” is not clear to me.
Author reply: Thank you for your valuable comments. We have changed total number of frequency to Number of 3’UTRs.
11. L242 the sentence is partly a repetition of those in line 223-224 and 224-225.
Author reply: Thank you for your valuable comments. We removed the sentence which was repeated twice in the manuscript.
12. L275, Figure 6: The Y-axis label is missing an "s" at the end of "target".
Author reply: Thank you for your valuable comments. Now we included “S” at the end of target in Figure 6.
13. L312. The sentence starting by “To normalize…” is a technical information that should not be included in the results section. Ditto for L344.
Author reply: Thank you for your comments. In both sections, we have changed and rewrote the sentences.
14. L382-384: I don’t understand the sentence and how the authors can conclude it supports the hypothesis that the genome of M. incognita is compact. Do the authors mean that the majority of the genes were expressed in all stages and only a very few were stage-specific, meaning that the transcriptome varies little between the different stages with a reduced repertoire of expressed genes? The authors should develop their idea and better explain this part.
Author reply: Thank you for your valuable comments. We have rephrased the sentences and put three new references.
15. L487: what is the “bad fraction”? Sequences with a low mean quality?. L488, I don’t understand what the authors mean by “low-quality bases…were removed”. Do they mean that low-quality bases are replaced by undefined nucleotides (N)? Or do they cut the 3’ extremity of reads when the quality is below a phred score of 20?
Author reply: Thank you very much for your comments. We rewrite the sentence to correct error like this. Raw reads sequences were trimmed by the parameters, quality trimming based on Phres quality scores (Q ≤ 20), adaptor trimming and minimum length discard (< 90 bp) using CLC genome cell (v. 4.0).
16. L491: the authors should provide information related to the reference genome they used.
Author reply: Thank you very much for your valuable comments to improve our current manuscript. We have included the reference genome id (ASM18041V1a) and incited reference.
17. L528. The sentence is not clear. What does the authors mean by “DESeq2 values”? Normalized abundances of reads? P-values?
Author reply: Thank you very much for your comments. We have changed to DeSeq2 and FPKM.
18. L544: there is no information about the PCR efficiency of each primer pair, whereas obtaining an efficiency close to 2 is a prerequisite to apply the 2-DDCq method. They could calculate the efficiency from data obtained with different template dilutions using the formula (E) = 10?1/slope as described by Pfaffl (2001) [1]. The authors should also provide stability values to justify their choice of the reference gene (p14, l3), using software such as geNorm or NormFinder. Pfaffl MW. A new mathematical model for relative quantification in real-time RT-PCR. Nucleic Acids Res. 2001;29:e45.
Author reply: Thank you very much for your valuable comments and reference article. During pilot experiments in our previous study (Ajjapala et al 2015, Disruption of prefoldin-2 protein synthesis in root-knot nematodes via host-mediated gene silencing efficiently reduces nematode numbers and thus protects plants and Vimalraj et al 2020, . Chitin biosynthesis inhibition of Meloidogyne incognita by RNAi-mediated gene silencing increases resistance to transgenic tobacco plants), we have validated the stability of beta actin (Accession number: BE225475.1) and 18sRNA (Accession number: U81578) across different development stages of nematodes and found that the expression of beta actin is stable across all stages. That is why we use the same (i.e. beta actin) in this experiment. We are already aware about the primer efficiency issue. However, it was impossible to examine the efficiency of 70 primer pairs. Nevertheless, as an alternative strategy, we have checked the uniqueness of all primer pairs using melting curve analyses in which all pairs showed a single peak only. However, we will consider these points in our future experiments. Thank you very much for your valuable comments again.
Round 2
Reviewer 2 Report
All the points I made in my previous review have been addressed by the authors, but I still don't know to what extent the methods/treatments used for sampling may affect the transcriptomes of the different stages.